# Reparameterization through Coverings and Topological Weight Priors

## Abstract

We generalise the reparameterization trick applied in variational autoencoders (VAEs) letting these have latent spaces of non-trivial topology – i.e. that of base manifolds covered with other ones, on which some technique for RT is available. That is possible since covering maps are measurable – moreover, in case of particular measure preservation property holding for the covering, one can establish an inequality on KL-divergence between pushforward (PF) densities on the base latent manifold, making the KL-term of VAE's ELBO analytically tractable, despite the topological non-triviality of the supporting latent manifold. Our development follows a route close but somewhat alternative to reparameterization on Lie groups, the latest proposal for which is to reparameterize PFs of normal densities from the Lie algebra – "through" the exponential map, seen by us as sometimes a particular case of what we propose to call reparameterization through a covering. Covering maps need not be global diffeomorphisms (although Lie-exp maps, in general, need not either, but, to date only smooth ones were considered in this context, to the best of our knowledge), which makes many non-trivial topologies tamable to our proposed technique, that we detail on a particular such example. We demonstrate the working of our approach by constructing a VAE with the latent space of Klein bottle (not a Lie group) topology, which we call KleinVAE, successfully learning an appropriate artificial dataset. We discuss potential applicability of such topology-informed generative models as weight priors in Bayesian learning, particularly for convolutional vision models, where said manifold was peculiarly shown to have some relevance.

## 1 Introduction

Natural data often possesses certain regularities, forcing it to concentrate near certain (latent) manifolds in the feature (representation) space – that is the statement of the manifold hypothesis. If some data is a priori known to possess such a manifold structure, this topological inductive bias should ideally be held in mind when modeling the data distribution – which can be done with e.g. variational autoencoders (VAEs), of all generative models. Constructing a VAE involves introducing a family of model conditional probability distributions of data latent features, called variational posteriors (VPs), the parameters of which can be inferred with low variance from input data, using the technique called the reparameterization trick (RT) (Kingma & Welling, 2013; Rezende et al., 2014).

As first introduced, RT implies that the VP is some input-data-conditioned normal distribution on Euclidean space of the latents, the topology of which is trivial (in homotopy theory sense: $\mathbb{R}^n$ is simply-connected, as its fundamental group is trivial), potentially unsuitable for modeling certain types of data (if the latent dimension is constrained). In this context, in recent years RT has been successfully generalized for latent spaces of various non-trivial topologies: e.g. ($n$-dim hyper-) spheres $S^n$ by Davidson et al. (2018) and Lie groups by Falorsi et al. (2019). Particular examples of the latter are: $n$-dimensional tori $T^n$ and, in fact, plain old (historically first to support RT) Euclidean spaces: with $\mathbb{R}^n$ forming both a Lie group and, at the same time, a Lie algebra of itself.

Inspired by these elegant developments, this paper proposes an alternative, somewhat more basically-topological view (more from the point-set topology and measure theory side of things) on RT in non-trivial topologies. As a model latent space, one can, in principle, consider any manifold (**base** of the **covering**) that can be **covered** with some other (**cover**) manifold, on the latter of

Figure 1: Two-sheet covering, see (2), $f_{2\to1} : T^2 \to \mathcal{K}$ of the Klein bottle $\mathcal{K}$ by a torus $T^2$. Each $y \in \mathcal{K}$ has 2 pre-images $x_1, x_2 \in T^2$.

which some technique for RT is available. In present work we consider the case when the **cover is the Euclidean space** $\mathbb{R}^n$, on which as a family of "source" VPs one can consider the family of normal distributions $\mathcal{N}(\mu, \Sigma)$ – particular probability measures (PMs) on $\mathbb{R}^n$. These can be **pushed-forward** (PF) onto the base **by the covering map** to make the family of "target", model VPs on that "target" base manifold.

The key fact (see Appendix) making this legit is that if both the base and the cover are measurable spaces, a covering between them is a **measurable mapping**: the PFs of source VPs from the cover constitute PMs on the base, so one can do probabilistic modeling on the "target" base manifold (with the cover being its source "proxy"). E.g. Monte-Carlo estimation of **expectations** (of measurable functions from the base manifold to real numbers – $\mathbb{R}$-valued random variables (RVs) on it) is done by just **sampling** from the source VP on the "proxy" cover, **mapping the sample point onto the base** with the covering map, and then averaging (the averaged function). This is required e.g. for computing the **evidence lower bound (ELBO)** of VAE models, since it has a form of expectation w.r.t. some probability measure on the latent space.

The claimed **contribution** of present work is that **reparameterization via coverings (RVC)** is also totally possible, due to said measurability of any covering map (it always being a local homeomorphism, so preserving the topology of measurable Borel sets), allowing to compute expectations in the above way: since the basic **idea of RT** is in the decomposition of the corresponding **(parameterized) stochastic (generative) map** into parts: 1) purely **stochastic sampling**, but of some **standard** (un-parameterized) RV; and then 2) re-**parameterization** of said standard RV's sample point with a **purely derministic reparameterization** map.

With present work we hope to show that said scheme can be performed on various non-trivial topologies as opposed to that of Euclidean spaces.

## 1.1 THE KLEIN BOTTLE & ITS TOPOLOGY IN IMAGE DATA

In present work we show how reparameterization can be done on a wide class of manifolds, that need not necessarily be Lie groups. As an minimal non-trivial example, we consider the Klein bottle, additionally motivated by its occurence in natural data. The Klein bottle $\mathcal{K}$ is a compact surface (manifold of dimension 2), that is non-orientable – and thus not a Lie group (since any Lie group is parallelizable). $\mathcal{K}$ can be realized as a unit square $[0, 1]^2$ with points on the edges identified:

$$(0, y) \equiv (1, y), \quad (x, 0) \equiv (1 - x, 1) \quad \forall x, y \in [0, 1], \tag{1}$$

so $\mathcal{K} \cong [0, 1]^2 / \equiv$, is a quotient of $[0, 1]^2$ by the above (1) equivalence relation. Notably, such realization of $\mathcal{K}$ can be covered by the 2-dimensional torus $T^2$ realized as a rectangle $[0, 2] \times [0, 1]$ with opposing points on the sides symmetrically identified (see (3) below), the covering map reads:

$$f_{2\to1}(x, y) = \begin{cases} (x, y) & \text{if} \quad x \le 1, \\ (x-1, -y) & \text{otherwise.} \end{cases} \tag{2}$$

This map (2) is a 2-sheet covering $f_{2\to1} : T^2 \to \mathcal{K}$, as any point $y \in \mathcal{K}$ has two pre-images $x_{1,2} = f^{-1}(y) \in T^2$ in the two parts (where $x \le 1$ or otherwise) of $T^2$. See Fig. 1.

Just as $\mathcal{K}$ can be two-sheet covered by $T^2$, the torus itself can be covered by $\mathbb{R}^2$, 2D Euclidean space – its quotient $\mathbb{R}^2 / \equiv$, by the following equivalence:

$$(x_1, y_1) \equiv (x_2, y_2) \quad \Leftrightarrow \quad \begin{cases} x_1 = x_2 \pmod 2 \\ y_1 = y_2 \pmod 1. \end{cases} \tag{3}$$

is a rectangle $[0, 2] \times [0, 1]$ with the above symmetric identification of edge points. This gives a map

$$f_{\mathbb{R}^2/\mathbb{Z}^2}(x, y) = ((x \mod 2), (y \mod 1)), \tag{4}$$

an infinite-sheet (any point on $T^2$ has infinitely many preimages on $\mathbb{R}^2$) covering $f_{\mathbb{R}^2/\mathbb{Z}^2} : \mathbb{R}^2 \to T^2$.

Combining maps (2) and (4) provides an infinite-sheet covering of $\mathcal{K}$ by $\mathbb{R}^2$:

$$f_{\mathbb{R}^2 \to \mathcal{K}} = f_{2 \to 1} \circ f_{\mathbb{R}^2/\mathbb{Z}^2}. \tag{5}$$

Note that, by construction, this covering is just, sheet-wise, an identical projection (albeit flipping the orientation on a half of all sheets) – the base $\mathcal{K}$ is covered by (infinitely many) sheets that are unit squares in $\mathbb{R}^2$, so an open ball $B_r(x)$ (Borel subset) neighborhood (of sufficiently small radius $r$) of any point $x \in \mathbb{R}^2$ is mapped onto an open ball of the same radius in $\mathcal{K}$ realized as $[0, 1]^2$ (modulo edgepoints equivalence). This makes the covering $f_{\mathbb{R}^2 \to \mathcal{K}}$ a very "nice" mapping – a measurable one – in fact, any covering is such, being a local homeomorphism.

Another reason we're interested in this particular manifold, $\mathcal{K}$ – is due to its appearance in natural data – which is, amuzingly, not limited to images: Klein bottle topology was also found by Martin et al. (2010); Stolz et al. (2020) in the energy landscape of conformations of cyclooctane molecules. In the limits of present work we only consider the appearence of $\mathcal{K}$ in natural images – see Discussion section for more details, at this point let us focus on purely topological side of things.

To model Klein bottle topology found in natural images, Love et al. (2023) propose to consider a certain subset of the Gabor filters (known in computer vision) – ones given by the following function:

$$F[\theta_1, \theta_2](x, y) = \sin(\theta_2) \, t_\theta(x, y) + \cos(\theta_2) \, Q\left(t_\theta(x, y)\right) \tag{6}$$

on the domain of a square $\{(x, y)\} = [-1, 1]^2$, parameterized by two angles, $\theta_1, \theta_2$, where $t_\theta(x, y) = \cos(\theta_1)\, x + \sin(\theta_1)\, y$ is projection onto a line specified by $\theta_1$, and $Q$ is a certain degree-2 Chebyshev polynomial: $Q(t) = 2t^2 - 1$. This function $F$ is clearly periodic in its parameter space $\Theta = \{\theta_1, \theta_2\}$, with period $[0, 2\pi]^2$ – but, moreover, satisfies the following condition:

$$F[\theta_1, \theta_2](x, y) = F[\theta_1 + \pi, -\theta_2](x, y), \tag{7}$$

making the fundamental domain in $\Theta$ not a torus $[0, 2\pi]^2$, but rather its subset $[0, \pi] \times [0, 2\pi]$ with Klein bottle topology. If $F$ is discretized on a grid, one gets what we refer to as Gabor-Klein filters.

To prove that Gabor-Klein filters indeed have the topology of $\mathcal{K}$, one can compute their persistent homology (PH) (Zomorodian & Carlsson, 2004; Weinberger, 2011). We do exactly that: by sampling $(\theta_1, \theta_2) \sim U[[0, 2\pi]^2]$ uniformly from the (double) fundamental region of $F$, we obtain a sample of 500 filters $3 \times 3$ filters and compute its PH with Ripser software library by Bauer (2021). These 500 filters are just real $3 \times 3$ matrices that we flatten turning them into a point cloud in $\mathbb{R}^{3 \times 3} = \mathbb{R}^9$. In this space, Euclidean distance is a natural metric, coinciding with Frobenius norm of difference of the matrices yet un-flattened – PH is homology of Vietoris-Rips filtration of its sub-level sets.

Persistence diagrams we obtain are shown on Fig. 2. We do not display 0-homology for visual clarity. Recall that reduced homology groups of the Klein bottle over $\mathbb{Z}$ are $H_0 \cong 0$, $H_1 \cong \mathbb{Z} \oplus \mathbb{Z}_2$ and $H_2 \cong 0$ (Hatcher, 2002). So, if computed over $\mathbb{Z}_2$ and $\mathbb{Z}_3$, they become respectively:

$$\begin{cases} H_1(\mathcal{K}; \mathbb{Z}_2) \cong \mathbb{Z}_2 \oplus \mathbb{Z}_2, \\ H_2(\mathcal{K}; \mathbb{Z}_2) \cong \mathbb{Z}_2, \end{cases} \quad \begin{cases} H_1(\mathcal{K}; \mathbb{Z}_3) \cong \mathbb{Z}_3, \\ H_2(\mathcal{K}; \mathbb{Z}_3) \cong 0. \end{cases} \tag{8}$$

This explains Fig. 2: over $\mathbb{Z}_2$, both components of $H_1$ are rightfully seen persisting, with one of them, originally $\mathbb{Z}$ over $\mathbb{Z}$, now reduced to $\mathbb{Z}_2$, yet still non-trivial. Over $\mathbb{Z}_3$, the other component of $H_1$, $\mathbb{Z}_2$ from the start over $\mathbb{Z}$ – disappears, hence not persisting. The "phantom" 2-homology of $\mathbb{Z}_2$ over $\mathbb{Z}_2$ rightfully disappears over $\mathbb{Z}_2$. Higher homology groups of $\mathcal{K}$ are trivial, since it's a 2-manifold. Aforementioned effects are due to what's called **torsion** of $\mathcal{K}$ being a non-orientable manifold. This raises the so-called field choice problem in computational homology (Obayashi & Yoshiwaki, 2019), that was in fact shown solvable by Boissonnat & Maria (2019) by effectively computing PH over many fields at once. These torsion effects make the topology of $\mathcal{K}$ (that can only be truly spotted by computing PH over $\mathbb{Z}_2$ and $\mathbb{Z}_3$ simultaneously) a notable test case for topological data analysis, as compared to other similarly low-dimensional manifolds.

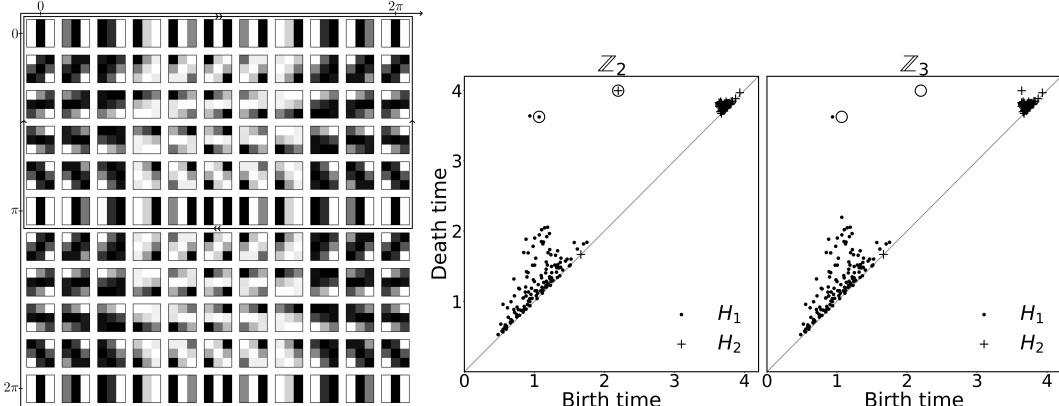

Figure 2: **Left:** Gabor-Klein filters = discretizations of function $F$ (6), with $\theta_1,\theta_2$ params, on a $3\times3$ grid. Upper half, $\theta_1 \leq \pi$, is the fund. region of $F$ in the parameter space $\Theta$, with Klein bottle topology. This is also an illustration of a 2-sheeted covering of $\mathcal{K}$ by a 2-torus: the whole square doubly covers the top half. **Right:** Persistence diagrams of 500 filters sampled from $U[\Theta]$: left is PH over $\mathbb{Z}_2$; right is over $\mathbb{Z}_3$. Persistent features that exist over $\mathbb{Z}_2$ and disappear over $\mathbb{Z}_3$ are circled.

## 2 REPARAMETERIZATION IN DIFFERENT TOPOLOGIES

Probabilistic generative modeling on manifolds is a broad and developing subject, reviewing it whole is beyond the scope of present paper. We only focus on variational autoencoder (VAE) models, with the latent space having some required non-trivial topology (as opposed to that of $\mathbb{R}^n$, Euclidean space). The reasons for such choice are as follows.

First: VAEs do not require an explicit parameterization of the latent manifold $\mathcal{Z}$, as a sufficiently expressive decoder (paired with a proper encoder) can itself learn a parameterization map embedding (if possible) $\mathcal{Z}$ back into the data generation space $\mathcal{X}$. All one needs is to **implicitly set latent topology** – that of $\mathcal{Z}$, that is: by introducing a reparameterizeable family of variational posterior (VP) probability measures (conditioned on data observed in $\mathcal{X}$) on $\mathcal{Z}$. Specifically, said VP family should permit: 1) sampling from the VP distribution on $\mathcal{Z}$ (to compute expectations of measurable functions – random variables – on $\mathcal{Z}$ with Monte-Carlo), 2) computing the **KL-divergence** between the VP and the prior distribution of latent parameters – coordinates on $\mathcal{Z}$ manifold. Both these computations should allow passage of their backward gradients w.r.t. parameters of VAE's layers, and, at that, the parameters of the VP distribution (as output of the encoder given sample points in $\mathcal{X}$ as input). Selecting VPs to be **push-forwards**, by some measurable map (covering being such), onto $\mathcal{Z}$ – allows these possibilities. **To sample** a point from a VP distribution being a PF by the covering map, one just samples from its **pullback** probability measure on the "proxy" cover of $\mathcal{Z}$, then projecting the sample point onto it, the base, with the covering map. By definition of pushforward, probability measures of all random events are "re-calculated" by the PF function correctly. As for reparameterizing this sampling map, the output of which is further fed to the decoder on a forward pass; and as for computing the KL-divergence between the VP and the prior defined on $\mathcal{Z}$ – both as PFs of their "proxies" (pullbacks) on the cover – see further. As we show, under an appropriate **measure-preservation property of the covering map**, an **analytic inequality on KL-divergence** arises between two "target" distributions on $\mathcal{Z}$, that are PFs by such covering – bounding it by the KL between their "proxy" pullbacks – from the right side, so **minimizing KL between pullbacks delivers minimizing it between the pushforwards** by such a covering map.

Second reason for consideration of present work being limited to VAEs is that despite, undoubtably, more expressive (density-based) topology-aware generative models were proposed, like e.g. normalizing flows by Rezende & Mohamed (2015) (which were generalized to have latent spaces of toric and spherical topology by Rezende et al. (2020), and potentially more general topologies by Brehmer & Cranmer (2020)), that are capable of learning multimodal distributions, VAEs can be seen as a basic building blocks for such models to figure out first. After all, the (un-) normalization map of flows can be seen as a more intricate development of the reparameterization map.

We see it that, with the proposed technique of reparameterization via coverings (RVC), detailed further, one has all the ingredients for a **topological VAE** (provided its model latent space has such topology that it admits a covering with another "proxy" manifold, whereon some technique for RT is already available). Such **TopoVAEs** should not be confused with **topological authoencoders**: TopoAEs as proposed by Moor et al. (2020) and RTD-AEs by Trofimov et al. (2023) – in these, the ambient latent space is Euclidean, into which the model learns to embed a point cloud of informative latent codes conditioned on observed input data – in such a way that the persistent homology these latent representations is close to that of input ones. Said closeness is measured with some (differentiable, at least almost everywhere, thus learnable) distance between the persistence diagrams, or as a norm of the diagram of a special cross-filtration (representation topology divergence of RTD-AEs). In contrast to this, the class of VAE models we propose initially has proper implicitly set latent topology. Also, making "vanilla" AEs data-generative is possible (Bengio et al., 2013), but is, sadly, an arguably underdeveloped direction of research; while presently proposed TopoVAEs are, architecturally, just VAEs, by-design generative models that are well understood, see Appendix.

## 2.1 EUCLIDEAN DENSITY REPARAMETERIZATION, SPHERES

The key to constructing a VAE is reparameterization (Kingma & Welling, 2013; Rezende et al., 2014). Consider it for the case when the latent space of a VAE is just the "backbone" 1D Euclidean space, the real line $\mathbb{R}$. The reparameterization (setting parameters $\mu$ and $\sigma$) transform on it reads:

$$z = f_{\mu,\sigma}(\varepsilon) = \mu + \sigma \cdot \varepsilon, \tag{9}$$

given as input a sample point $\varepsilon \sim \mathcal{N}(0,1)$ of a standard normal random variable (RV), $f_{\mu,\sigma}$ is a (deterministic) map that outputs such a sample point of a normal RV $z \sim \mathcal{N}(\mu,\sigma)$ with parameters $\mu, \sigma$. To construct a Topo-VAE with a topologically non-trivial latent space $\mathcal{Z}$, one needs to generalize (9) (or some version of it) to work on this space.

Note that (9) exploits the the linear (**affine**, to be precise) structure of $\mathbb{R}$. The real variable $\varepsilon$ that is input to $f_{\mu,\sigma}$ can be seen as a 1D vector, lying in $\mathbb{R}$. With $\mathbb{R}$ forming a vector space (over itsel, $\mathbb{R}$, as a base field), vectors in it can be **added** and **multiplied by a number** from the base field, the latter equivalent to adding a vector to itself some real number of times. This, in fact, is also a manifestation of 1D Euclidean space $\mathbb{R}$ being a **Lie group** $G$, with (9) giving the affine group action on this manifold: a combination of affine maps is again an affine map, and it depends on its parameters $\mu, \sigma$ smoothly. Notably, $G$ also serves as a **Lie algebra** $\mathfrak{g}$ for itself – being a vector space as above explained. The (Lie) **exponential map** from $\mathfrak{g}$ to $G$ here is just the identity map.

The above described view is exactly what was proposed (generalized to other non-trivial examples) by Falorsi et al. (2019) as reparameterization on manifolds that are Lie groups. It should be noted, however, that Lie group structure is only one of many possible structures a manifold can be equipped with (if not to say it's arguably one of the "nicest", hence most restrictive, structures) and one can potentially exploit those other ones to design reparameterization tricks.

For example, if the latent manifold $\mathcal{Z}$ is Riemannian (equipped with a metric), one can do reparameterization by sampling random walks (diffusion) on it, as Rey et al. (2019) propose: the intuition behind this is that in the limit, random walks converge to some form of Riemannian normal law with density at point $z$ proportional to $\exp(-d^2(\mu,z))$. There is a lot of progress in this direction of generative modeling on manifolds (see also (Gemici et al., 2016; Yu et al., 2025; Kalatzis et al., 2021) for more flow-based approaches), but present work follows a different, more measure-theoretic, path (with a VAE as a basic vehicle): we do not explicitly require a Riemannian structure of the manifold (except perhaps for it having a volume form so one can talk of Riemannian measure on it).

Before fully diving into Lie group, let us briefly mention a very interesting case of (hyper-) spherical topologies – separately from Lie groups, since, exotically, not all spheres are Lie groups (not even all of them admit a weaker, topological group structure (Megia, 2007)). Reparameterization can be generalized to spheres of arbitrary dimension (Davidson et al., 2018) exploiting other structures: e.g. using a probabilistic technique of reparameterization through accept-reject sampling on spheres was proposed by Naesseth et al. (2017), thus only appealing to the compactness of spheres. We envision an alternative approach: computing implicit reparameterization gradients proposed by Figurnov et al. (2018) "through" smooth automorphisms of spheres. A very similar technique for spherical normalizing flows via Möbius transformations group was proposed by Rezende et al. (2020).

## 2.2 LIE GROUP DENSITY REPARAMETERIZATION

Our work was most inspired by reparameterization generalized to the case of the latent manifold being a Lie group by Falorsi et al. (2018; 2019). For a brief reminder on Lie groups, see Appendix.

A Lie group $G$ is a **smooth manifold** that also possesses a group structure. For an example, imagine a unit circle, which is a 1-sphere $S^1$, embedded in Euclidean plane $\mathbb{R}^2$ with Cartesian coordinates, centered at its zero origin. Any point on $S^1$ can be parameterized with an angle $\varphi$ that one counts wrapping around the circle counterclockwise starting from point with coordinates $(1, 0)$. A line tangent to the circle at this point $(1, 0)$ (identity group action element of $G$) can be seen as a vector space $\mathbb{R}$ – this is the Lie algebra $\mathfrak{g}$ of $G$. One can say that the angles $\varphi$ parameterizing points on $S^1$ "live" in this line (in the algebra $\mathfrak{g}$ being a vector space) – angles can be added and multiplied by a real number. There is a smooth map from $\mathfrak{g}$ to $G$, called the exponential map, which in this example, given an angle $\varphi \in \mathfrak{g}$, returns a point on $S^1$ (in $G$) with coordinates $(\cos \varphi, \sin \varphi)$ in the plane.

To work with **distributions on the sphere** $S^1$, it being a Lie group $G$, one can consider distributions on the very familiar $\mathfrak{g} \cong \mathbb{R}$ – e.g. normal distributions. Any normal probability density $\mathcal{N}(\mu, \sigma)$ on $\mathfrak{g} \cong \mathbb{R}$ can be pushed onto $G \cong S^1$ by the **exponential map** – its pushforward is called wrapped normal distribution (on the circle). The idea of reparameterization on Lie groups is that one can reparameterize distributions on Lie algebras of Lie groups (this works at least for compact connected ones (Falorsi et al., 2018; 2019)) – which are then **pushed forward** onto the group by the exponential map. So, the measurable space of the Lie algebra $\mathfrak{g}$ serves as a "proxy" of that of the Lie group $G$ it corresponds to: instead of working with probability densities on $G$, one can work with their $\mathfrak{g}$-supported "proxies". Lie algebras of many Lie groups are isomorphic to Euclidean spaces $\mathbb{R}^n$, with normal distributions as one possible choice of localised yet expressive distribution models on such.

This **reparameterization through the exp map** onto Lie groups is, in fact, also a particular case of what we call **reparameterization through the coveging** map, since the above described exponential map from $\mathfrak{g} \cong \mathbb{R}$ to $G \cong S^1$ also delivers an infinite-sheet covering map! Although not always, Lie-exp maps sometimes also serve as **universal covering** maps $\mathfrak{g} \to G$ (when $G$ is connected and abelian). These two different yet similar classes of maps: coverings of $G$-s and Lie-exp maps $\mathfrak{g} \to G$ thus provide two ways to map non-trivial topologies in a way that allows reparameterization of probability densities on these. An important distinction between these two constructions (that are otherwise very similar in this case of $G \cong S^1$) is that Lie-exp maps are more often expected to be (almost-) **global diffeomorphisms** (for $\mathfrak{g} \cong \mathbb{R} \to G \cong S^1$ the above described Lie-exp is such, due to $\sin(\cdot)$ and $\cos(\cdot)$ functions being smooth except at one infinity point), while covering maps need not at all – not expected to be more than **local homeomorphisms**, they seem less restrictive, and thus perhaps more practically helpful, as if the covering map is **not smooth** at some points – one could not care less, since sometimes this set of points of derivative discontinuity is of **measure zero**.

## 2.3 COVERED SPACES: CIRCLE, KLEIN BOTTLE

We outline a method to reparameterize probability measures on certain manifolds – namely, ones that can be covered with other ones (on which one can do reparameterization) in a certain way (such that the **measure is not-increased sheet-wise** by the covering). This is possible since any covering is a measurable map. We do so on a particular example – the Klein bottle, motivated by the mentioned appearance of this peculiar manifold in natural data. For a memo on coverings, see Appendix.

### 2.3.1 RVC ON THE CIRCLE OF $S^1$

Topologically, $S^1$ can be realized as a quotient $\mathbb{R}/\mathbb{Z}$ of the real line $\mathbb{R}$ by the integer lattice $\mathbb{Z}$ – by the following equivalence relation:

$$x \equiv y \quad \Leftrightarrow \quad x = y \pmod 1, \tag{10}$$

so the circle $S^1$ is isomorphic to an interval $[0, 1]$ with endpoints identified, $0 \equiv 1$. The **map**

$$f(x) = (x \mod 1) \tag{11}$$

(taking the fractional part of a number) thus provides a **covering** $f : \mathbb{R} \to [0, 1]$ of the **base** interval with $\mathbb{R}$ (the **cover**). This is an infinite-sheet covering: every point $x \in [0, 1]$ has (countably) infinitely many **pre-images** $f^{-1}(x)$ in the cover.

The real line $\mathbb{R}$ with a **Borel sigma algebra** $\mathcal{B}(\mathbb{R})$ on it, $(\mathbb{R}, \mathcal{B}(\mathbb{R}))$ – is a **measurable space** that can be equipped with a "natural" measure – Lebesgue measure $\lambda$, with respect to which one can integrate measurable functions (compute expectations of RVs). Normal distributions $\mathcal{N}(\mu, \sigma)$ form a family of probability distributions (on $\mathbb{R}$) of very special interest – these are often used as VPs.

The distribution of a **normal RV** $X \sim \mathcal{N}(\mu, \sigma)$ is a **probability measure** $P_X$ on $\mathbb{R}$ that is absolutely continuous w.r.t. $\lambda$, so it has a probability density function (PDF) $p_X(x)$, the Gaussian:

$$dP_X = p_X \, d\lambda = (2\pi\sigma^2)^{-1/2} e^{-(x-\mu)^2/2\sigma^2} \, d\lambda. \tag{12}$$

The covering (11) $f : \mathbb{R} \to [0,1]$ is a **measurable map** from $(\mathbb{R}, \mathcal{B}(\mathbb{R}), \lambda)$ to $[0,1]$ (with a Borel sigma-algebra on it), so one can consider the pushforward (PF) $f_* P_X$ of $P_X$ – a measure on $[0,1]$. **Pushforward measure** of a small neighborhood $B = B_\varepsilon(y)$ of any point $y \in [0,1]$ is thus given by

$$f_* P_X(B) = \sum_{\{A_i\} = f^{-1}(B)} P_X(A_i), \tag{13}$$

where $A_i$ are small open neighborhoods of (infinitely many) points $\{x_i\} = f^{-1}(y)$ – pre-images of $y$. In fact, sheet-wise (restricted to any sheet $\mathcal{X}_i \subset \mathbb{R}$, $f$ is a bijection), the covering map $f$ is just an identical projection, so Lebesgue measure $\lambda$ is pushed unchanged from every sheet:

$$d(f_* P_X)(y) = \sum_i p_X(x_i) \, d\lambda = \Big( \sum_i p_X(x_i) \Big) \, d\lambda, \tag{14}$$

so it makes sense to talk about the **pushforward density** $f_* p_X$ (Radon-Nikodym derivative of $f_* P_X$ w.r.t. $\lambda$), which, at any point $y$, is given by the above equation 14 infinite sum (that converges due to properties of the Gaussian) of normal densities $p_X$ at pre-images $\{x_i\} = f^{-1}(y)$. This distribution on the circle $S^1 \cong [0,1)$ is referred to as **wrapped normal distribution** – it inherits many properties of the normal on $\mathbb{R}$: in particular, it concentrates (the more the smaller the value of $\sigma$) around its single mode at $f(\mu)$, so it is naturally used to model spherical RVs in directional statistics.

A very important consequence of $f$ **preserving measure** $\lambda$, sheet-wise (constrained on any sheet), is the following statement. Consider two probability distributions on $\mathbb{R}$ – with densities $q$ and $p$. If these are pushed onto $[0,1]$ by $f$ (denote pushforwards $q^* = f_* q$ and $p^* = f^* p$) and one considers Kullback–Leibler (KL-) divergence between the PFs is given by:

$$\mathrm{KL}(q^* \,||\, p^*) = \int_{[0,1]} q^* \, \log \frac{q^*}{p^*} \, d\lambda \tag{15}$$

where both $q^*$ and $p^*$ at any point $y \in [0,1]$ equal to sums $q^*(y) = \sum_i q(x_i) = \sum_i q_i$, same for $p^*(y) = \sum_i p_i$ of pullback densities at pre-images $x_i$ of $y$. Then, by the **log sum inequality** (following from Jensen's inequality), the integrand, and hence the integral, in (15) is bounded from above by the KL-divergence:

$$\mathrm{KL}(q^* \,||\, p^*) \leq \int_{[0,1]} \sum_i q_i \, \log \frac{q_i}{p_i} \, d\lambda = \mathrm{KL}(q \,||\, p) \tag{16}$$

between the original (pullback) densities $q$ and $p$ on $\mathbb{R}$. The above is because, if one permutes the sum and the integral in (16), one just gets a sum of integrals over all the sheets of the covering, which, due to **sheet-wise preservation of measure** $\lambda$, just equals the integral over the whole $\mathbb{R}$.

### 2.3.2 Inequality on KL under coverings, RVC on the Klein bottle

Central to our proposed reparameterization technique is the result of an **inequality on KL**-divergence under covering maps between measurable spaces. In particular case of a **sheet-wise measure-preserving** map $f(x) = (x \bmod 1)$ covering each point of $[0,1]$ with (measurably) infinitely-many pre-images (those of any point indexed $i$ below) from $\mathbb{R}$, the inequality (16) reads:

$$\int_{[0,1]} q^* \, \log \frac{q^*}{p^*} \, d\lambda = \int_{[0,1]} \sum_i q_i^* \, \log \frac{q_i^*}{p_i^*} \, d\lambda \leq \int_{[0,1]} \sum_i q_i \, \log \frac{q_i}{p_i} \, d\lambda. \tag{17}$$

If said sheet-wise (Lebesgue) measure $\lambda$ preservation property is relaxed to **sheet-wise non-increase of measure** – generic measure $m$, $B(x)$ for a Borel "ball" set containing point $x$:

$$\forall x_i \ : \ f_* m(B(x_i)) \leq m(f^{-1}(B(x_i))), \tag{18}$$

one trivially – **proof** follows from 1) linearity (of summation w.r.t. the pre-logarithm density multiple, with any point-wise re-scaling of measures not affecting the densities ratio in the logarithm), 2) the fact since the push-forwards are on the left hand side of both (17) and (18) and 3) the **log sum inequality** – obtains present paper's self-contained mathematical **result**, namely:

**Proposition 1** *Given $\mathcal{X}$ and $\mathcal{Y}$ – measurable spaces; $P$ and $Q$ – probability measures on $\mathcal{X}$; $p$ and $q$ – their corresponding probability densities – Radon-Nikodym derivatives w.r.t. some ambient measure $m$ on $\mathcal{X}$; and a **sheet-wise measure-non-increasing covering map** $f : \mathcal{X} \to \mathcal{Y}$; $p^* \, dm$ and $q^* \, dm$ – pushforwards of $p \, dm$ and $p \, dm$ onto $\mathcal{Y}$ by $f$; it **holds true** that **KL-divergence is non-increasing under the covering** $f$: $KL(p^* \,||\, q^*) \leq KL(p \,||\, q)$.*

With that established, one can consider the case of Klein bottle $\mathcal{K}$ topology. As described in the introduction, $\mathcal{K}$ can be realized as a unit square $[0,1]^2$ with points on the sides appropriately identified. It can be 2-sheet covered (2) by a 2-torus, $T^2$, realized as a rectangle $[0,2] \times [0,1]$ with endpoints identified symmetrically as in (3). Denote said covering map $f_{2\to1}$. Note that $f_{2\to1}$ is a sheet-wise identical projection map (albeit flipping the orientation on half of the sheets) – so if $T^2 \cong [0,2] \times [0,1]$ is equipped with Lebesgue measure $\lambda$, inherited from $\mathbb{R}^2$, then on each sheet $\lambda$ is preserved by $f_{2\to1}$. The torus, in turn, can be realized as a quotient of $\mathbb{R}^2$ by $\mathbb{Z} \times 2\mathbb{Z}$ – this provides a (infinite-sheet) covering, denoted $f_{\mathbb{R}^2/\mathbb{Z}^2}$, which is also, sheet-wise, just a truly identical projection preserving $\lambda$. So $\mathcal{K} \cong [0,1]^2$ is covered with $\mathbb{R}^2$ with a map

$$f = f_{2\to1} \circ f_{\mathbb{R}^2/\mathbb{Z}^2} : \mathbb{R}^2 \to \mathcal{K} \tag{19}$$

that is **sheet-wise measure-preserving** for $\lambda$. With that, one can introduce RVs on $\mathcal{K}$, by introducing RVs on $\mathbb{R}^2$ and then pushing their distributions onto $\mathcal{K}$ by $f$.

Thus to construct a **TopoVAE** with its **latent space topology** that of $\mathcal{K}$, one can pick the **PFs of normal distributions** $\mathcal{N}(\mu, \Sigma)$ from on $\mathbb{R}^2$ onto $\mathcal{K}$ by $f$ as a **family of VPs**. PFs of $\mathcal{N}(\mu, \Sigma)$ onto $T^2$ by $f_{\mathbb{R}^2/\mathbb{Z}^2}$ are known in (directional) statistics as **multivariate wrapped normal distributions**. The unimodality of $\mathcal{N}(\mu, \Sigma)$-s may break when these are pushed onto $\mathcal{K}$ by $f$, but at least in the limit of small variance, when $\mathcal{N}(\mu, \Sigma)$ is close to a delta-measure at $\mu$, its PF by $f$ will also concentrate around one point on $\mathcal{K}$. Such divergence of VP from unimodality can be controlled with the prior.

So consider two normal probability measures $P$ and $Q$ on $\mathbb{R}^2$, with parameters $\theta_p = (\mu_p, \Sigma_p)$ and $\theta_q = (\mu_q, \Sigma_q)$ and denote their PFs onto $\mathcal{K}$ by $f$ with $P^*$ and $Q^*$. Since measure is preserved sheet-wise by $f$, $P^*$ and $Q^*$ have PDFs in $\mathcal{K}$ – $p^*$ and $q^*$, which, at any point on $\mathcal{K}$, equal sums of PDFs of $P$ and $Q$ on $\mathbb{R}^2$, $p$ and $q$, at all pre-images of that point.

With that one constructs a particular **TopoVAE**, that we'll refer to as **KleinVAE**, as follows:

- latent space: $\mathcal{Z} = [0,1]^2 \cong \mathcal{K}$,
- PDF of **VP** (of latent $z$ given observed $x$): $q^*(z|x)$ – PF of $\mathcal{N}(\mu_q, \Sigma_q)$ by $f$ (see Eq. 19),
- PDF of **prior** (of latent $z$, unconditioned by $x$): $p^*(z)$ – PF of $\mathcal{N}(\mu_p, \Sigma_p)$ by $f$ (see Eq. 19).

On a **forward pass**, the **encoder**, given an input batch of data $x$, outputs the parameters $\theta_q$ of the VP $q^*$. By definition of PF (or by so-called LOTUS, see Appendix), sampling a RV $z \sim f_* q$ can be done by first sampling from $q$ and then applying $f$ to the result. This random sample $z \sim q^*_{\theta_q}$ is then passed to the **decoder** that outputs the parameters of $p(x|z)$ – the **likelihood of reconstruction** $x$ given latent $z$. In turn, the gradient of **evidence lower bound (ELBO)**:

$$\mathrm{ELBO}(W) = \mathbb{E}_{z \sim q^*_{\theta_q(W_{\mathrm{enc}})}} \, p_{W_{\mathrm{dec}}}(x|z) - \mathrm{KL}(q^*_{\theta_q(W_{\mathrm{enc}})} \,||\, p^*_{\theta_p}) \tag{20}$$

w.r.t. $W = (W_{\mathrm{enc}}, W_{\mathrm{dec}})$ – the parameters of the VAE network's layers is easily **backward passed**. Due to sheet-wise preservation of measure by $f$, **KL-divergence** between $q^*$ and $p^*$ on $\mathcal{Z} = \mathcal{K}$ is **bounded from above** by KL between $q$ and $p$ on $\mathbb{R}^2$ – so the negative KL term (20) in **ELBO is bounded from below** by

$$-\mathrm{KL}_{\mathcal{K}}(q^*_{\theta_q} \,||\, p^*_{\theta_p}) \geq -\mathrm{KL}_{\mathbb{R}^2}(q_{\theta_q} \,||\, p_{\theta_p}), \tag{21}$$

– KL-div between two bivariate Gaussians $q_{\theta_q}$ and $p_{\theta_p}$, for which a well-known **analytic formula** exists. So maximizing ELBO, equivalent to that if (20) with the KL-term changed for this $\mathrm{KL}_{\mathbb{R}^2}$ – is computationally advantageous (more **tractable**), mitigating **Monte-Carlo** estimation of KL.

So what's left is to define how to **backward-pass** the gradient (w.r.t. $\theta_q$) through sampling (in the first, reconstruction loss, term of ELBO):

$$z \sim q_{\theta_q}^* = f_* q_{\theta_q} = f_* \text{PDF}_{\mathcal{N}(\mu_q, \Sigma_q)} \tag{22}$$

– this stochastic map is **decomposed** into sampling a standard RV $\varepsilon \sim \mathcal{N}(0, I)$, and then then reparameterizing it

$$z = f(\mu_q + \Sigma_q^{1/2} \varepsilon). \tag{23}$$

This provides gradient passage through **reparameterization via covering**, since the $f$ map is a covering map, being part of the reparameterization map (23). With the covering being (19) simply a sheet-wise identical projection (albeit changing the orientation on half of the sheets), **differentiating** it is trivial, and can be safely delegated to automatic differentiation – as $f$ is only discontinuous at the boundaries of the sheets, which are a set of Lebesgue **measure zero** in $\mathbb{R}^2$. Practically it is thus enough to implement $f$ with e.g. differentiable `torch.where` and `torch.remainder` functions of Pytorch library by Paszke et al. (2019).

## 3   CONCLUSION

In present work, we show how one can do the reparameterization trick of variational inference on various topologically non-trivial spaces. This allows to use these as latent spaces of topology-aware generative models, as we exemplify by constructing a Topo-VAE with the latent space having the topological structure of the Klein bottle. This is novel since, to date, reparameterization for such non-trivial topologies has been explicitly generalized, of comparably non-trivial cases – only to Lie groups. As opposed to that, our proposed technique of reparameterization via coverings, that is more grounded in measure theory (we state a Proposition on non-increase of KL-divergence between pushforwards of probability measures by a measure-non-increasing covering map between two measurable topological spaces) – does not expect global differentiability of the reparameterization map, allowing to have an arbitrarily complicated set of its discontinuities, provided it has measure zero. We construct such a TopoVAE with latent Klein bottle topology and report its performance on an appropriate artificial dataset. We also speculate on applicability of such topology-aware generative models as model weight priors in Bayesian learning, providing potentially helpful topological inductive bias.

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

# A APPENDIX I: EXPERIMENT, KLEINVAE

## A.1 ARTIFICIAL CIRCLES DATASET

As a proof of validity of our method, we implement a VAE with the latent space having the topology of the Klein bottle, that we call KleinVAE. We experimented with learning the distribution of Gabor-Klein filters (see Fig. 2) with it, but found it not so easy with shallow autoencoders with standard layers – possibly due to the non-linear oscillating nature of these filters. That is why we provide a demo of KleinVAE on some other artificial data.

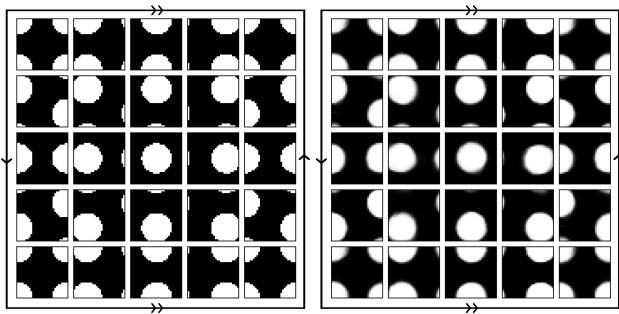

Figure 3: Klein-Circles dataset. **Left**: samples of original data – circles on the Klein bottle with centers at different positions. **Right**: reconstructions of corresponding images with KleinVAE. White = intensity equals 1, black = zero.

We generate small images ($30 \times 30$ pixels) of a circle of radius 0.3 on a unit square. Boundary points of the square are identified in such a way that it has topology of $\mathcal{K}$. With circle center at uniformly random positions, this forms our **Klein-Circles dataset**. We generate $100.000 (= 10^5)$ such images.

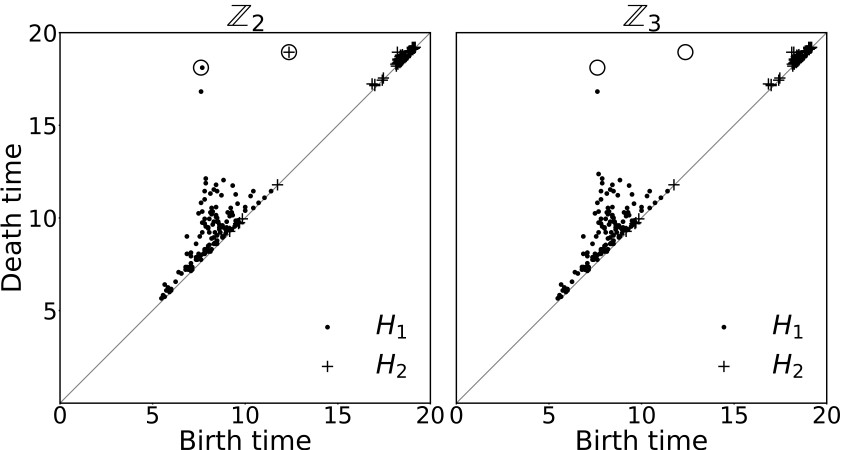

Figure 4: Persistence diagrams of 500 images decoded by KleinVAE from images with uniformly random circle center positions.

We train a KleinVAE with the encoder made of fully-connected layers of sizes $30 \times 30 = 900 \rightarrow 1024 \rightarrow 512 \rightarrow 128 \rightarrow 32 \rightarrow 5$ (so the latent space has dimension 5, since on it we predict 5 parameters of the VP: 2 components of the $\mu$ vector and 3 components of the lower-triangular scale matrix, that, squared, provides the $2 \times 2$ covariance matrix of the VP), and the decoder with similar ones but in reverse sequence, with Leaky-ReLU nonlinearity. Batch size is set to 1024, training is done with Adam optimizer Kingma & Ba (2014) for 50 epochs with initial learning rate lr $= 10^{-3}$ with LR scheduler `ReduceLROnPlateau` with reduction factor of 0.99. KL term in ELBO is set to have a weight of $10^{-2}$. The prior is set to a (pushforward onto $\mathcal{K}$ of) normal with $\mu = (0.5, 0.5)$ (center of the square) and $\Sigma = \text{diag}(0.1, 0.1)$.

We compute persistent homology of reconstructions of a sample of $500$ images with uniformly random circle center positions - persistence diagrams can be seen on Fig. 4. These suggest that KleinVAE successfully learns the latent structure of the Klein bottle.

## A.2    ABLATION STUDY

As an ablation study of utility of such topological inductive bias in VAEs, we trained several VAE models: 1) ordinary VAE with latent spaces of different dimensions (2D, 3D, 4D), 2) VAE with the latent space being a Torus $T^2$, which we refer to as the TorusVAE, and 3) VAE with the latent space being the Klein bottle $K$, which we refer to as the KleinVAE.

In order to estimate how good the models were in the learning an embedding of proper $\mathcal{K}$ topology into the data space, we compute the persistent homology of the reconstructed hold-out subset of the validation set. Due to computational cost of this procedure, we randomly selected $500$ points from the validation set as the hold-out subset. We used the Python's `Ripser.py` Tralie et al. (2018) interface for `Ripser` Bauer (2021) for persistent homology computations, and computed persistent homology in dimensions 0, 1 and 2.

To numerically compare the persistent diagrams of the original hold-out set with its reconstruction, we computed the bottleneck distance $d_B$ between the diagrams. The bottleneck distance is defined in terms of the bottleneck cost $c_M$

$$c(M) = \max\left\{ \sup_{(p,q)\in M} \|p - q\|_\infty, \sup_{s \in P \sqcup Q \text{ unmatched}} \frac{|s_y - s_x|}{2} \right\},$$

where $(p,q) \in P \times Q$ is called the matched pair, if $\forall p \in P$ there exists at most one $q \in Q$ and $\forall q \in Q$ there exists at most one $p \in P$. An element $s \in P$ (resp. $Q$) is unmatched, if it has no pair in $Q$ (resp. $P$).

The bottleneck distance $d_B$ between two diagrams $P$ and $Q$ is the smallest bottleneck cost achieved by partial matchings between them

$$d_b(P,Q) = \inf_{M:P \leftrightarrow Q} c(M).$$

The bottleneck distance is computed independently for each homology dimension. To create a single aggregated metric, we report the $\ell_2$-norm of the vectors of these distances

$$\|d_B(P,Q)\|_2 = \left( \sum_{h=0}^n d_B(P_h, Q_h)^2 \right)^{1/2},$$

where $P_h$ and $Q_h$ are the persistence diagrams for the original data and its reconstruction in homology dimension $H$ respectively.

We trained these models within the same setup for 200 epochs, with the encoder and decoder made as fully-connected networks with 1 hidden layer of dimension $64$. Batch size was set to $1024$, learning rate was set to $0.01$, optimized with Adam (Kingma & Ba, 2014) with the `ReduceLROnPlateau` LR Scheduler with factor $0.99$. We used $\mathrm{Leaky-ReLU}$ as a non-linear activation function. The weight of KL term in ELBO calculation was set to $0.001$.

From Figures 5 and 6 that, although the topological-fidelity metricd (bottleneck distances) of "vanilla" (plain Euclidean) VAEs seem better; the reconstruction metrics of "topology-aware" models (toric and Klein-bottle ones) are better – since the covering map does not alter the ambient Lebesgue measure, the part of ELBO that is (negative) KL between Normals on the plane and the other topologies – is identical, so the difference is in reconstruction lossess only. We have a **hypothesis** as to why "topology-unaware" models did somewhat better in reconstructing it, as measured by homological bottleneck distance: these models have "more room" in the Euclidean space (that is "chopped of" by the covering map) to tesselate it with "remembered" latent codes: to check this, we also compute the variance of the latent points. Plot (b) on 5 does not violate our hypothesis. We envision a detailed explanation of observed effects as our future work.

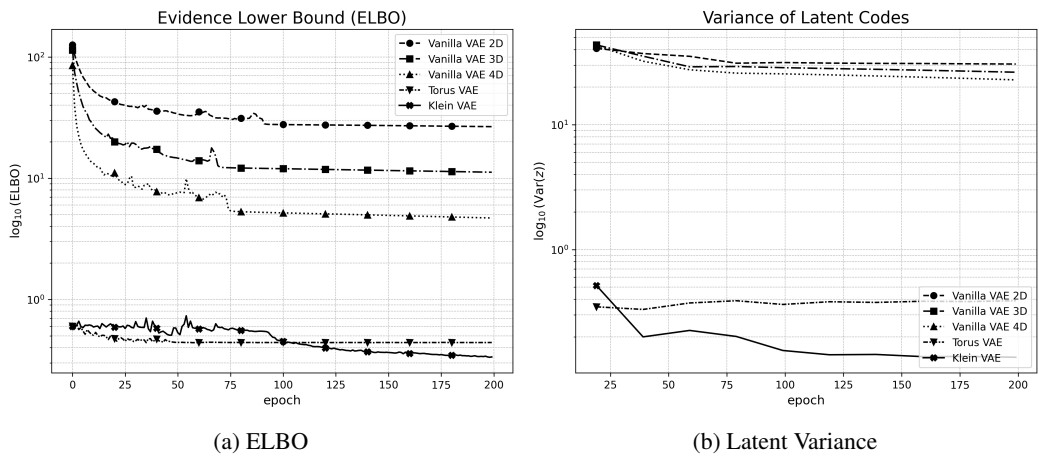

(a) ELBO                    (b) Latent Variance

Figure 5: Training dynamics and topological metrics for all models. (a) Evidence Lower Bound, (b) Latent code variance

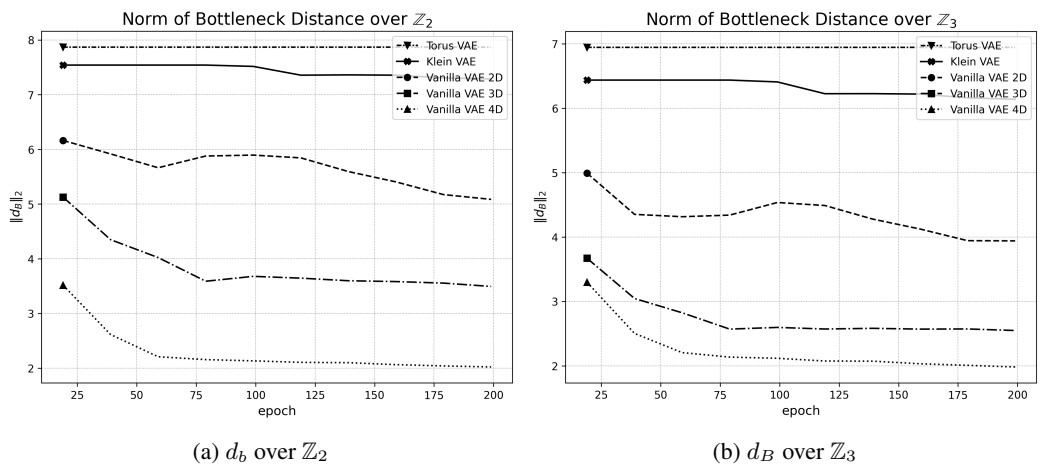

(a) $d_b$ over $\mathbb{Z}_2$                    (b) $d_B$ over $\mathbb{Z}_3$

Figure 6: (a-b) Total bottleneck distance over $\mathbb{Z}_2$ and $\mathbb{Z}_3$.

# B  APPENDIX II: OUTLOOK

With the main claimed contribution of present work being a generalization of RT – with reparameterization via coverings shown possible, advocated to be used in generative modeling (not limited to VAEs – seen as basic latent models that can be used as building blocks for more expressive ones to figure out first), we're not merely interested in this technique per se. We hope to highlight connections seen between several directions of research: gen. modeling, Bayesian learning, and topological data analysis (TDA) – by proposing an exemplar construction of such a topology-aware (manifold-supported) generative models, that can be used as a prior (in the sense of Bayesian inference) on the parameters of some other model, thus providing it with relevant topological inductive bias.

## B.1  BAYESIAN LEARNING

Bayesian inference in machine learning, or just Bayesian learning (BL) – is a motivation of present work. BL is developing since at least 90s (Neal, 1996; Neapolitan, 2003), and is still relevant in the modern age of deep learning (DL) (Wang & Yeung, 2020; Fortuin, 2022).

Any model processing data $X$ is specified by its parameter $\theta$ – if the model is a neural network (NN), $\theta$ contains the weights of its layers. In BL, instead of considering one optimal parameter value given

training data $X$, one considers

$$p(\theta|X) = \frac{p(X|\theta)\,p(\theta)}{p(X)} \tag{24}$$

the whole posterior probability distribution $p(\theta|X)$ of possible parameter $\theta$ values given observed data $X$. From Bayes' theorem (24), it follows that $p(\theta|X)$ is proportional to the likelihood $p(X|\theta)$ of observed data $X$ given a certain value of $\theta$ – times the **prior** $p(\theta)$ probability of parameter $\theta$ taking a certain value, independent of observed data. With a prior $p(\theta)$ one can incorporate all of knowledge of what values $\theta$ should or should not likely take – into the model. Typically, such Bayesian models are trained with Monte-Carlo (MC) techniques, for which it is enough to be able to **sample** random values $\theta \sim p(\theta)$ **from the prior**.

In principle, incorporating informative prior beliefs into models improves their performance and generalization ability (if the prior is "strong" enough compared to the amount of training data available). We were particularly motivated by weight priors for vision models: it was shown by Pearce et al. (2020) that imposing priors on the structure of the filters of convolutional layers of CNNs (the original LeNet (LeCun et al., 1989), VGG (Simonyan & Zisserman, 2014), ResNet (He et al., 2016))) improves their performance (faster training to reach same accuracy) and generalization ability (higher entropy of predictions of an untrained CNN initialized only from the prior); see also (Fortuin et al., 2021). Said imposed structure is, in fact, well-known – filters are sampled from the family of **Gabor filters** (Marĉelja, 1980) (rather than from uncorrelated Gaussian noise components), that were widely used in CV before the success of CNNs. This "revival" of Gabor filter usage – as weight initialization for CNNs – happened before: (Luan et al., 2018; Alekseev & Bobe, 2019), but it was Pearce et al. (2020) who properly framed this as Bayesian learning.

Further on, it was proposed by Atanov et al. (2018) to learn (with a generative model, e.g. a VAE) such distributions of useful NN weights (CNN filters) from a NN trained on one dataset – to transfer those as priors for NNs operating on other datasets – this was called **Deep Weight Prior** (DWP), following the introduction of Deep Image Prior (DIP) by Ulyanov et al. (2018).

Said works strongly motivated present work, which advocates paying more attention to topological structures when doing probabilistic modeling, as, in particular, do Jin et al. (2024). As described further, one such a topological structure for CNN weights in particular has been known for some time, so we propose to combine the flexibility of priors learned with a generative model (e.g. a VAE) with such "**topological inductive bias**": knowing which exact manifold the prior distribution is in fact supported on. We propose to refer to such priors as **Topological Weight Priors**.

### B.2 Priors for Topological Deep Learning: Klein bottle in vision

Another side of this, seemingly, single direction of research – is **topological deep learning (TDL)**. Its motivating premise is that data possesses certain topological structure (e.g due to restrictions of symmetry) to account for when designing models processing such data (Carlsson, 2009).

One particular observation that inspired much development in the field called **topological data analysis (TDA)** – is that such structure is found in natural images (Lee et al., 2003): if one considers grayscale images of natural objects at a small scale – i.e. (high contrast) patches of e.g. $3 \times 3$ pixels, the distribution of such patches would not fill the entire feature space $\mathbb{R}^9$, but would rather concentrate near a certain (embedded) manifold of dimension 2. Using the construction of persistent homology (Barannikov, 1994; Robins, 1999; Edelsbrunner et al., 2002; Zomorodian & Carlsson, 2004), the vehicle of TDA, it was shown by Carlsson et al. (2008) that said manifold, in fact, has a topological structure very similar to that of the Klein bottle $\mathcal{K}$.

It was proposed by Perea & Carlsson (2014) to utilize this knowledge to build low-parameter dictionaries of natural textures. More recently, it was proposed by Gabrielsson & Carlsson (2018); Carlsson & Gabrielsson (2020); Love et al. (2023) to directly incorporate said topological structure into CNNs – authors refer to these as **Topological CNNs** (TCNNs). The idea is clear: by design (LeCun et al., 1989), convolutional filters of CNNs are meant to detect repeating motifs of small image patches, activating most on patches they are most co-aligned with – so if certain structure is present in typical image patches, a similar, dual structure should emerge in the filters of a trained CNN, this is exactly what Gabrielsson & Carlsson (2018) observed. These filters are, in fact, a certain subset of the mentioned Gabor filters – to which we refer to as **Gabor-Klein filters**.

The proposal of Love et al. (2023) with TCNNs is thus to incorporate these Gabor-Klein filters into (at least, early) layers of CNNs by sampling these from their (Klein bottle) topology. That is done essentially by discretizing this topology into a graph and sampling from a (discrete) distribution on it. In contrast, **our proposal** is to learn a smooth distribution of informative filters on a smooth representation of such a priori known topologies – with an appropriate genererative model.

## C  APPENDIX III: BACKGROUND

### C.1  COVERINGS

Recall Hatcher (2002) that a covering is a continuous map (can be thought of, and so referred to, as a projection, albeit not necessarily linear)

$$c : \mathcal{X} \to \mathcal{Y} \tag{25}$$

between topological spaces (where $\mathcal{X}$ is said to be the **covering space**, or simply the **cover**; and $\mathcal{Y}$ is said to be the **base**), such that every point $y \in \mathcal{Y}$ has an open neighborhood $U_y$ such that

$$c^{-1}(U_y) = \coprod_{d \in D_y} S_d \tag{26}$$

its pre-image is a discrete space $D_y$ – a direct sum (disjoint union) of **sheets** $S_d$ – open subsets of $\mathcal{X}$, such that projection from every sheet

$$c|_{S_d} : S_d \to U_y \tag{27}$$

is a homeomorphism. The discrete set $c^{-1}(y)$ of pre-images of any point $y \in \mathcal{Y}$ is called the **fiber** of $x$. We will only consider the case when the base $\mathcal{Y}$ is connected, so the covering map is surjective, and the cardinality of $D_y$ is the same $\forall y \in \mathcal{Y}$ – this value is called the **degree** of the covering. One may refer to coverings of degree $n$ as $n$-sheet or $n$-fold. This value need not be finite – we will consider countably infinite degree coverings.

### C.1.1  UNIVERSAL COVERINGS

Two coverings of the same base space, $c : \mathcal{X} \to \mathcal{Y}$ and $\tilde{c} : \tilde{\mathcal{X}} \to \mathcal{Y}$, with simply connected covers can be shown to be equivalent – i.e. there exists a uniquely determined homeomorphism $h : \mathcal{X} \to \tilde{\mathcal{X}}$ such that the diagram

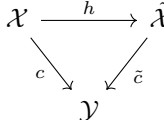

commutes. So in this case the covering $(\mathcal{X}, c)$ is determined uniquely up to said equivalence, and is called the **universal covering** of $\mathcal{Y}$. If the base $\mathcal{Y}$ is locally simply connected (i.e., any neighborhood $U_y$ of any point $y \in \mathcal{Y}$ contains a "smaller" neighborhood $V_y \subset U_y$ which is simply connected), then a universal covering $c : \mathcal{X} \to \mathcal{Y}$ exists, and can be defined constructively. Any topological manifold (topological space that is locally Euclidean) is locally simply connected and thus admits a universal covering.

### C.1.2  EXAMPLES OF COVERINGS

The 1-sphere $S^1$ represented as a circle of unimodular complex numbers

$$S^1 \simeq \{z \in \mathbb{C} \mid |z| = 1\} \tag{28}$$

is $n$-sheet covered with a $n \in \mathbb{N}$ power map

$$f : z \to z^n \tag{29}$$

– any point in the circle has $n$ pre-images.

The 1-sphere $S^1$ represented as a unit circle centered at the origin in Euclidean plane is covered by the real line $\mathbb{R}$ with a map

$$f_S : t \to (\cos(t),\ \sin(t)) \tag{30}$$

– which is equivalent to first factorizing $\mathbb{R}$

$$f_{./2\pi\mathbb{Z}} : t \to t \mod 2\pi \tag{31}$$

by the $2\pi\mathbb{Z}$-lattice (which provides a covering of $[0, 2\pi)$, where the endpoints $0$ and $2\pi$ are identified – with $\mathbb{R}$), and then applying $f_S$ (which, on $[0, 2\pi)$ is already injective, so the important part is taking the quotient (31)). With that $\mathbb{R}$ is, in fact, made to be the **universal cover** of $S^1$.

An $n$-torus, being a product of 1-spheres,

$$T^n \simeq \underbrace{S^1 \times \cdots \times S^1}_{n \text{ copies}} \tag{32}$$

is thus universally covered with the product $\mathbb{R}^n$, by factorizing each coordinate with (31).

## C.2 MEASURE AND PROBABILITY

Any topological space $\mathcal{X}$ can be equipped with a **Borel sigma-algebra** $\mathcal{B}(\mathcal{X})$ – the smallest sigma-algebra containing all sets obtainable with the operations of countable union, countable intersection and relative complement to open (or equivalently – closed) subsets of $\mathcal{X}$.

A topological space with its sigma-algebra $(\mathcal{X}, \mathcal{A})$ is called a **measurable space** if it can be equipped with a **measure** – a function $m : \mathcal{A} \to \mathbb{R} \cup \{\pm\infty\}$ from $\mathcal{A}$ to the extended real line such that:

1. **non-negative**: $\forall A \in \mathcal{A} :\ m(A) \geq 0$,
2. $m(\emptyset) = 0$ where $\emptyset$ is the empty set,
3. **countably-additive**: $m\left(\bigcup_{i=1}^{\infty} A_i\right) = \sum_{i=1}^{\infty} m(A_i)$ for any $\{A_i\}_{i=1}^{\infty}$ – countable collection of pairwise disjoint sets in $\mathcal{A}$,

On the real line $\mathbb{R}$ (and, by extension, on any $n$-dimensional Euclidean space $\mathbb{R}^n$) we will only consider the "natural" measure – **Lebesgue measure** $\lambda$.

A measurable space $(\mathcal{X}, \mathcal{A})$ with a measure $m$ on it – $(\mathcal{X}, \mathcal{A}, m)$ is called a **measure space**. In probability theory, one considers **probability spaces** – i.e. measure spaces $(\Omega, \mathcal{F}, P)$, such that

1. $\Omega$ is a non-empty set of elementary events (outcomes), called the **sample space**,
2. $\mathcal{F}$ is a sigma-algebra on it,
3. $P$ is a certain measure on $\mathcal{F}$, called a **probability measure**: it assigns a measure equal to 1 to the whole sample space, $P(\Omega) = 1$.

### C.2.1 PUSHFORWARD

Given two measurable spaces $(\mathcal{X}, \mathcal{A}_\mathcal{X})$ and $(\mathcal{Y}, \mathcal{A}_\mathcal{Y})$, a **measurable mapping** between them

$$f : \mathcal{X} \to \mathcal{Y} \tag{33}$$

– i.e. such that the pre-image of any $A_\mathcal{Y} \in \mathcal{A}_\mathcal{Y}$ is an element of $\mathcal{A}_\mathcal{X}$:

$$f^{-1}(A_\mathcal{Y}) = \{x \in \mathcal{X} \mid f(x) \in A_\mathcal{Y}\} \in \mathcal{A}_\mathcal{X}, \tag{34}$$

and a measure $m_\mathcal{X} : \mathcal{A}_\mathcal{X} \to [0, +\infty]$, a "**pushforward** of $m_\mathcal{X}$ by $f$", denoted by $f_* m_\mathcal{X}$, given by

$$f_* m_\mathcal{X}(A_\mathcal{Y}) = m_\mathcal{X}(f^{-1}(A_\mathcal{Y})) \quad \forall A_\mathcal{Y} \in \mathcal{A}_\mathcal{Y} \tag{35}$$

is a measure on $(\mathcal{Y}, \mathcal{A}_\mathcal{Y})$.

### C.2.2 RANDOM VARIABLES

Given a probability space $(\Omega, \mathcal{F}, P)$, a **random variable** (RV) $X$, taking values in $\mathcal{X}$ ($(\mathcal{X}, \mathcal{A}_\mathcal{X})$ is a measurable space) is a measurable mapping

$$X : \Omega \to \mathcal{X}. \tag{36}$$

Assume $(\mathcal{X}, \mathcal{A}_\mathcal{X})$ is equipped with some "reference" measure $m_\mathcal{X}$. Oftentimes the codomain $\mathcal{X}$ is taken to be $\mathbb{R}^n$, with $\mathcal{A}_\mathcal{X}$ being the Borel sigma-algebra $\mathcal{B}(\mathbb{R}^n)$ on it, and the reference measure is Lebesgue measure $\lambda$ on $\mathcal{B}(\mathbb{R}^n)$.

If the pushforward $X_*P$ is absolutely continuous w.r.t. the reference measure $m_\mathcal{X}$ (required to be so-called sigma-finite), denoted $X_*P \ll m_\mathcal{X}$ (i.e., for any $m_\mathcal{X}$-measurable set $A \in \mathcal{A}_\mathcal{X}$, $m_\mathcal{X}(A) = 0$ implies $X_*P(A) = 0$), then the Radon-Nikodym theorem states the existence of a function $p_X : \mathcal{X} \to [0, \infty)$, which is $m_\mathcal{X}$-measurable, such that:

$$X_*P(A) = \int_A p_X \, dm_\mathcal{X} \tag{37}$$

– called **probability density function (PDF)** of RV $X$. The integral here is in Lebesgue integral, which can be defined in certain cases for arbitrary topological spaces. It thus measures probabilities of events $A \in \mathcal{A}_\mathcal{X}$ – that $X$ takes certain values:

$$\Pr(X \in A) = \int_{X^{-1}A} dP = \int_A p_X \, dm_\mathcal{X}. \tag{38}$$

We will refer to $P_X$ (measure on $\mathcal{A}_\mathcal{X}$) as **probability distribution** of RV $X$, meaning that:

$$dP_X = d(X_*P) = p_X \, dm_\mathcal{X}. \tag{39}$$

Standard notation for "random variable $X$ follows distribution $P_X$" reads:

$$X \sim P_X \quad \text{or equivalently} \quad X \sim p_X. \tag{40}$$

### C.2.3 EXPECTATIONS

For a RV $X \sim P_X$ one can compute **expectations** of measurable functions $f : \mathcal{X} \to \mathbb{R}$ (where $\mathbb{R}$ is equipped with Borel sigma-algebra $\mathcal{B}(\mathbb{R})$ and Lebesgue measure $\lambda$)

$$\mathbb{E}_{P_X} f(X) = \int_\Omega f \circ X \, dP = \int_\mathcal{X} f \, dP_X, \tag{41}$$

where the subscript of $\mathbb{E}_{P_X}$ (somewhat redundant in this case) means we are averaging $f(X)$ with $P_X$. This (41) is sometimes referred to as the **law of the unconscious statistician (LOTUS)**.

Now consider a sequence of measurable mappings:

$$\Omega \xrightarrow{X} \mathcal{X} \xrightarrow{f} \mathcal{Y} \xrightarrow{g} \mathbb{R} \tag{42}$$

– so $X \sim P_X$ is a RV on $\mathcal{X}$, $Y = f(X)$ is a RV on $\mathcal{Y}$, and we consider a measurable function $g$ of $Y$. If

$$Y \sim f_*P_X \tag{43}$$

– the distribution of $Y$ is a pushforward of $P_X$, then, if $g \circ f$ is integrable w.r.t. $P_X$, it holds true (see Theorem 3.6.1 in Bogachev (2007)) that

$$\int_\mathcal{Y} g \, d(f_*P_X) = \int_\mathcal{X} g \circ f \, dP_X \tag{44}$$

iff $g \circ f$ is integrable w.r.t. $P_X$. Note that the mapping $f$ is ***not required to be injective*** – if so, to compute the contribution of a certain point $y \in \mathcal{Y}$ (more precisely, its "small" open neighborhood $U_y \supset y$) to the LHS of (44), one has, by definition of pushforward (35), to ***sum over all pre-images***:

$$\int_{U_y} g \, d(f_*P_X) = \int_{f^{-1}(U_y)} g \circ f \, dP_X, \tag{45}$$

which can be nontrivial. Luckily, though, if one is only looking to compute the expectation of $g(Y)$, where $Y \sim f_*P_X$, then, by LOTUS (41),

$$\mathbb{E}_{f_*P_X} g(Y) = \int_\mathcal{X} g \circ f \, dP_X \tag{46}$$

– if integrating over the whole $\mathcal{X}$ is somehow easier, one can just do that.

### C.2.4 MONTE-CARLO

In practice, computing expectations of (measurable) functions $f : \mathcal{X} \to \mathbb{R}$ of a RV $X \sim P_X$ is often done with Monte-Carlo (MC) methods: given access to a generator of (independent) random samples

$$X_n = \{x_i\}_{i=1}^n, \quad \text{where} \quad x_i \sim P_X, \tag{47}$$

the expectation is approximated with large ($n \gg 1$) sample average:

$$\mathbb{E}_{P_X} f(X) \approx \frac{1}{n} \sum_{i=1}^n f(x_i). \tag{48}$$

## C.3 LIE GROUPS

A **Lie group** $G$ is a smooth (real, finite-dimensional) manifold that also possesses a structure of a group – that is, group operations (multiplication and inversion), are smooth maps (see e.g. Kirillov (2008) for a detailed introduction). Formally, it is enough to require that:

$$(x, y) \to x^{-1} y \tag{49}$$

is a smooth mapping from $G \times G$ to $G$. Many examples of Lie groups are subgroups of the **general linear group** $\mathrm{GL}(n, \mathbb{R})$ of invertible $n \times n$ matrices (with matrix multiplication as group product).

### C.3.1 LIE ALGEBRA

The **Lie algebra** $\mathfrak{g}$ of an $n$-dimensional Lie group $G$ is its tangent space ($G$ is a smooth manifold) at $1_G$ – the identity group element. $\mathfrak{g}$ is an $n$-dimensional vector space of "infinitesimal generators", from which elements of $G$ can be obtained with the exponential map.

### C.3.2 EXPONENTIAL MAP

The **exponential map**

$$\exp(\cdot) : \mathfrak{g} \to G \tag{50}$$

maps elements of the algebra $\mathfrak{g}$ to elements of the group $G$ – e.g. $\exp(0) = 1_G$. The exponential map is *surjective* if $G$ is compact and connected, but, in general, ***not injective*** – so the inverse of $\exp(\cdot)$, the **logarithm map**, $\log(\cdot)$ – is in general multi-valued.

### C.3.3 ADJOINT

The Lie algebra $\mathfrak{g}$ of a Lie group $G$ is naturally equipped with a binary operation called the **Lie bracket** $[\cdot, \cdot] : \mathfrak{g} \times \mathfrak{g} \to \mathfrak{g}$, which is bilinear, and $\forall x \in \mathfrak{g}, [x, x] = 0$. With it, one defines the **adjoint representation** of $x \in \mathfrak{g}$:

$$\mathrm{ad}_x(y) = [x, y]. \tag{51}$$

### C.3.4 EXAMPLES

An illustrative example of a Lie group is the 1-sphere $S^1$ – consider it embedded to $\mathbb{R}^2$ as a unit circle centered at the origin. Any point on the circle is specified by the angle $\varphi$ – it can be obtained by taking the vector $(1, 0)^T$ and acting on it with a matrix:

$$R_\varphi = \begin{pmatrix} \cos \varphi & -\sin \varphi \\ \sin \varphi & \cos \phi \end{pmatrix}. \tag{52}$$

This is the matrix representation of $SO(2)$ – special orthogonal group (group of rotations of the 2D plane), acting on the circle. Note that

$$R_\varphi = \exp \left[ \varphi \begin{pmatrix} 0 & -1 \\ 1 & 0 \end{pmatrix} \right] \tag{53}$$

– any group element $R_\varphi$ is given by the matrix exponential (which is a smooth map)

$$\exp(X) = \sum_{k=0}^n \frac{X^k}{k!} \tag{54}$$

of the single basis element of the Lie algebra $\mathfrak{g}$ spanned by this matrix of "infinitesimal" rotation – including $R_0 = I_2$, $2 \times 2$ identity matrix which is the group identity element $1_G$. Thus (53) provides the exponential map from the Lie algebra $\mathfrak{g} \simeq \mathbb{R}$, where the rotation angle $\varphi$ "lives" – one can add and subtract rotation angles; to the Lie group $G = SO(2) \simeq S^1$.

Note that this Lie exponential map (54) $R_\varphi : \mathbb{R} \to SO(2) \simeq S^1$ is smooth in its argument angle $\varphi$ – everywhere except at one infinity point. Modulo the latter, it can be thus said to be a global diffeomorphism. The angle $\varphi$ in it, however, is defined modulo $2\pi$ – at that, $R_\varphi$ is also an infinite-sheet covering of $S^1$ with $\mathbb{R}$. But such a covering can be constructed even simpler: as the following quotient-ing map (same as (31):

$$f_{\mathbb{R}/2\pi\mathbb{Z}} : t \to (t \mod 2\pi) \tag{55}$$

– that covers with $\mathbb{R}$ an interval $[0, 2\pi]$ with endpoints identified. Note that this covering map is, sheet-wise, just an identity projection, so its Jacobian clearly equals one except at the endpoints where this map is continuous but not smooth, as opposed to the matrix exponential map (52). Jacobian of the latter everywhere from on $\mathbb{R}$, by the Pythagorean theorem, also equals one, so a pushforward of standard Borel measure $\mathbb{R}$ (Lebesgue measure $\lambda$ in this case) by either of these maps is very simple – one just needs to sum Lebesgue measures of pre-images of any subset of the covering base $S^1$, without any non-unitary weights obtained from the Jacobian.

### C.3.5 Haar measures

The **Haar measure** $\mu_{\text{Haar}}$ is a canonical way to equip a Lie group $G$ (or generally any locally compact Hausdorff topological group) with the measure structure. It is defined as a regular Borel measure that is left-invariant under the group action: for any Borel set $S \subseteq G$ and for any $g \in G$, we have $\mu_{\text{Haar}}(gS) = \mu_{\text{Haar}}(S)$.

Haar measures always exist and are unique up to a positive scaling factor; when $G$ is compact, the Haar measure can be normalized to be a probability measure.

### C.4 Variational Autoencoders

Variational autoencoders (VAEs) (introduced in Kingma & Welling (2013), see also Kingma et al. (2019) for a detailed introduction) are a family of generative models: given a training sample $X_n$ of data (points in the data space $\mathcal{X}$), a VAE learns to map small subsamples (batches) of it onto the space $\mathcal{Z}$ of "latent" parameters describing the distribution this subsample came from – so similar datapoints can be sampled from it. This can be understood from the perspective of Bayesian inference: Bayes' theorem states

$$p(z|x) = \frac{p(x|z)\,p(z)}{p(x)} \tag{56}$$

that the **posterior** probability $p(z|x)$ of (latent) parameter $z$ taking a certain value for a datapoint $x$ – is given by the **likelihood** $p(x|z)$ of observing this datapoint given $z$; times the **prior** probability $p(z)$ of such $z$; divided by the **evidence** $p(x)$ of the datapoint $x$. The latter is a source of a problem:

$$p(x) = \int_{\mathcal{Z}} p(x|z)\,p(z)\,dz \tag{57}$$

might be intractable (especially if the dimension of $\mathcal{Z}$ is high). This is mitigated with the technique of **variational inference** – instead of computing the exact posterior $p(z|x)$, one aims to find an approximation $q_{\theta_q}(z|x)$ – thus called the **variational posterior (VP)** – such that

$$\text{KL}(q_{\theta_q}(z|x)\,||\,p(z|x)) \to \min_{\theta_q} \tag{58}$$

the Kullback–Leibler divergence

$$\text{KL}(q(z)\,||\,p(z)) = \int_{\mathcal{Z}} q(z)\,\log\frac{q(z)}{p(z)}\,dz \tag{59}$$

between them is minimized. It can be shown (by Jensen inequality) that KL-divergence is non-negative, and zero iff $q = p$ (as measures), so minimizing it makes distributions similar. Minimization in (58) is done in a sense that $q$ is chosen from some family of probability distributions (called the

**variational family**), a representative of which is chosen by setting the parameters $\theta_q$ equal to some value. Since (58) still involves the true posterior $p(z|x)$, one uses the following observation:

$$\log p(x) = \mathbb{E}_{z \sim q} \log \frac{p(x, z)}{q(z|x)} + \mathrm{KL}(q(z|x) \,||\, p(z|x)) \tag{60}$$

that (log-)evidence, which does not depend on $z$ and thus on parameters $\theta_q$ of $q$, is bounded from below (since KL-divergence is always $\geq 0$) with the first term in (60), which is thus called **evidence lower bound (ELBO)**. So instead of minimizing KL between $q$ and $p(z|x)$, one can as well maximize ELBO, which can be rewritten (with (56)) as:

$$\mathrm{ELBO}(\theta_q) = \mathbb{E}_{z \sim q} \log p(x|z) - \mathrm{KL}(q(z|x) \,||\, p(z)) \tag{61}$$

Notably, both terms of (61) are expectations w.r.t. $q$ – so, if one can sample $z \sim q$, these can be computed with Monte-Carlo (48).

In VAEs, $q(z|x)$ and $p(x|z)$ are given by neural networks. The **encoder** network, parameterized with $W_{\mathrm{enc}}$, given a batch of input data $x$, outputs parameters $\theta_q$

$$\theta_q = \mathrm{encoder}_{W_{\mathrm{enc}}}(x) \tag{62}$$

of the VP $q$, which is typically chosen to come from the family of normal distributions $\mathcal{N}(\mu, \Sigma)$ (with $\theta_q = (\mu_q, \Sigma_q)$) on the latent space $\mathcal{Z}$, which is Euclidean space $\mathbb{R}^D$. If the prior $p(z)$ is also chosen to be a normal distribution with parameters $\theta_{\mathrm{prior}}$, then the KL term of ELBO (61) admits a closed-form analytic expression in terms of $\theta_q$ and $\theta_{\mathrm{prior}}$.

The **decoder** network, parameterized with $W_{\mathrm{dec}}$, given a (random) sample point

$$z \sim q_{\theta_q(x)} \tag{63}$$

($q$ parameterized with $\theta_q$ output by the encoder), outputs

$$\theta_{p(x|z)} = \mathrm{decoder}_{W_{\mathrm{dec}}}(z) \tag{64}$$

the parameters $\theta_{p(x|z)}$ of the likelihood $p(x|z)$ of the datapoint $x$. If $x$ is e.g. a binary image, the likelihood $p(x_i|z)$ of its pixels $x_i$ can be set to Bernoulli distribution, with $\theta_{p(x_i|z)} \in [0, 1]$ being its parameter.

A VAE model is thus trained by optimizing ELBO w.r.t. the parameters $W = (W_{\mathrm{enc}}, W_{\mathrm{dec}})$ of the network – typically by some version of stochastic gradient descent (SGD). Since sampling (63) a point $z$ from the VP $q_{\theta_q}$ is part of computing the optimization objective function (the "forward pass" of a VAE), to train a VAE one has to compute the gradient of this (stochastic) mapping w.r.t. $\theta_q$ (and then "backward pass" the gradient to $W_{\mathrm{enc}}$). This can be done with the technique known as the **reparameterization trick** Kingma & Welling (2013); Rezende et al. (2014): if the VP $q_{\theta_q}$ is chosen to be a normal distribution, then to obtain a random sample $z \sim q = \mathcal{N}(\mu_q, \Sigma_q)$, one can

1. sample from the standard normal $\varepsilon \sim \mathcal{N}(0, I)$
2. apply a loc-scale transform $z = r_{\theta_q}(\varepsilon) = \mu + \Sigma^{1/2}\, \varepsilon$

(where $\Sigma^{1/2}$ can be found from the covariance matrix $\Sigma$ with e.g. Cholesky decomposition). With that one decomposes the stochastic mapping (63) into deterministic $r_{\theta_q}$ reparameterization mapping, and stochastic sampling mapping $z \sim \mathcal{N}(0, \Sigma)$ that has no parameters to optimize.

