# OpenReview forum: "Reparameterization through Coverings and Topological Weight Priors"
_ICLR.cc/2026/Conference — ICLR 2026 Conference Withdrawn Submission_

### Official Review · Reviewer_RUB6 · 2025-10-25

**Soundness:** 2
**Presentation:** 2
**Contribution:** 2
**Rating:** 4
**Confidence:** 2

**Summary:**

Reparameterization trick is a key component in variational autoencoders, allowing gradient to back propagate to the parameters. The paper proposes a generalization of the reparameterization trick to latent spaces with complex topology. By utiliziing the covering maps, one can pushforward a tractable density, e.g. a multivariate normal, to the manifold. While previous works have explored similar idea in Lie groups, the paper generalizes upon those by enabling the reparameterization trick with latent spaces that are not a Lie group, e.g. the Klein bottle. Discussions on potential applications are provided.

**Strengths:**

1. The problem of learning an VAE with a latent space of complex topology is interesting.
2. Sufficient mathematical analysis are provided.

**Weaknesses:**

1. The empirical results are provided only in the appendix, and appears to at a proof-of-concept level.
2. In general, I find the paper a bit hard to read as a machine learning paper.

**Questions:**

I am not that familiar with the research topic.

As mentioned above in weaknesses, I find the empirical experiments lacking in the main paper and the paper hard to read.

The paper is written in a somewhat unconventional manner: bold texts are often employed, and the experiments are delayed entirely to the appendix. The texts are also confusing to read at times, e.g. Line 378: "one trivially - proof follows from...".

---

> ### Author Response · Authors · 2025-12-01
>
> Dear Reviewer,
> Thank you very much for your time and effort, and attention to our modest work!
> We've decided to withdraw it from ICLR, to further improve it.
> Again, thank you very much for your review, we highly appreciate all aspects of it!
> With all respect,
> Authors

---

### Official Review · Reviewer_ETSM · 2025-10-26

**Soundness:** 3
**Presentation:** 1
**Contribution:** 2
**Rating:** 2
**Confidence:** 5

**Summary:**

The paper introduces a method of incorporating topological constraints to latent spaces of VAEs (or more generally any situation where one variationaly learns a distribution on a nontrivial topology). The idea is to use a covering map from a flat space onto the latent space with the nontrivial topology. During training or inference one takes the pushforward via the covering map, to obtain and sample from distributions on the latent space. Given that the original distributions over the flat space (which the authors take to be the Lie algebra of the group of transformations on the Klein bottle manifold in their chief example, which coincides with the Lie group itself) can be parametrized differentiably, and given that the covering map is differentiable, the setup allows for variational inference to use gradient information to learn a distribution on the Klein bottle manifold, or any other similar nontrivial topology which can be covered by a simple enough space.

The author(s) focus on the Klein bottle topology, and use methods of persistent homology to demonstrate that various surfaces do indeed have the homological hallmarks of Klein bottle topology. In training with the standard VAE objective, the authors note that since the non-pushforwarded distributions (gaussians) on the flat covering space upper bound the KL-divergence of their pushforwards, they may as well replace the regularization term in the ELBO objective with the KL divergence of the distributions on the covering space.

The applications are relegated to the appendix, where various VAEs (with various latent space topologies) are trained on a synthetic dataset that is designed to have the Klein bottle topology. The results are then compared to some extent.

**Strengths:**

The author(s) do bring a lot of mathematical tools on the table, and show their connection in a setup where that could have potential applications and give practitioners novel tools. The authors do bring forward the idea of covering spaces to parametrize distributions on nontrivial topologies. This idea can be found in Falorsi et al which the authors correctly attribute as a source of their inspiration, but there the emphasis was on the Lie group and lie algebra structure and the exponential map, which does bring more tools to the table (such as the use of harmonic analysis tools, see figure 4.1 in Explorations in Homeomorphic Variational Auto-Encoding) on top of the existence of the exponential map as a canonical/distinguished covering map. But perhaps those tools weren't strictly necessary, and this work shows that one can still get the setup for VAE-on-nontrivial-topology without needing anything more than a covering map from a simple enough covering space.

**Weaknesses:**

I encourage the author(s) to resubmit the paper to another conference taking note of the following points to improve both the presentation and the focus of their investigation. The summary of all the points below is that I would encourage the author(s) to find a convincing use case for the mathematical tools they present from a machine learning point of view.
The main text of the paper does not include any experiments, the computational aspect is relegated to the appendix, and even there it is not sufficient to demonstrate the utility of using the Klein bottle topology (or any other nonstandard topology for that matter) as the latent space of a VAE, or again, in any other way. I do respect the generality of what is presented in the main text and understand that it is not only restricted to a VAE latent space topology but there is no convincing use case in this paper. Otherwise, the whole setup is exciting. I think a much more carefully designed experiment needs to be done to showcase how a KleinVAE is able to perform a certain representation task better and that this is precisely due to its topology.

I would argue that the inclusion and the presentation of advanced mathematical tools is done for the love of the tools only. This is something I have sympathy towards, since I too find these mathematical tools beautiful. However in the context of machine learning it is also important to include a tool only if it serves a purpose for the construction or the understanding of the model.
* For example in line 140, the author(s) start with the phrase ``to prove that''. Two problems with this. Persistence diagrams cannot prove that Gabor-Klein filters have the topology of $\mathcal{K}$. Secondly, this is totally a futile pursuit as equation (7) itself shows that the explicit parametrization of Gabor filters has the Klein bottle topology. There is no need to use advanced cohomology results (the authors are using implicitly the characterization of surfaces without boundary as one of sphere/connected sum of g tori(genus g orientable surface)/ connected sum of k real projective plane, as Klein bottle is the only one among them that satisfies the given homology criterion). Furthermore persistent homology is a non-exact tool, errors arise due to the approximation of random sampling (notice for example in figure 2 right there are $H_2$ homologies popping up over $Z_3$, which shouldn't happen). This would be fine if it was the only way of showing that the given surface has the topology fo $\mathcal{K}$ but in this case it is apparent from  the analytical form of the parametrizing equations of the Gabor filters as it satisfies (7).
One case I can see where persistent homology calculations like the ones in the paper being of legitimate use is if they are applied to datapoints, for example in the latent space of a vanilla euclidean VAE, and we want to show that these points are lying on a manifold that has the Klein bottle topology. Otherwise it is much too heavy of a machinery to imprecisely demonstrate a fact we already know easily by other means.

To quote Andre Weil (from the foreword of his book Number Theory) talking about why he chose to not treat Galois cohomology in his book  ``For me to develop such an approach systematically would have meant loading a great deal of unnecessary machinery on a ship which seemed well equipped for this particular voyage; instead of making it more seaworthy, it might have sunk it." I am afraid this current paper sank under the weight of all the mathematical tools the authors chose to include.

As another example, despite my personal love for Lie groups, I have to say that the examples that are investigated do not warrant a Lie group section. The only reason they are mentioned is because of the exponential map $\exp: \mathfrak{g}\to G$ as a covering map and when that map (for the Klein bottle) is the identity since $\mathfrak{g} \cong G \cong R^2$. There is no mention of the group operation, because there was no need for it, which is fine, it only goes to show that the structure of a Lie group wasn't fully utilized. This is mentioned in passing on line 295. So if it is the case then why is the text "diving fully into Lie groups" as stated in line 262?

Section 2.3 spends a page and a half on two instances of how KL-divergence becomes smaller under pushforwards instead of citing the Data Processing Inequality (see, e.g. Polyanskiy and Wu, 2014, Thm 6.2) stating that "Let g be a measurable function, then KL(g∗P,g∗Q) ≤KL(P,Q)," The purpose of this section is only to justify the use of the KL-divergence in the covering space instead of the KL-divergence in the KleinBottle topology space for the regularizer, so it is not necessary to devote this much space to it. Instead the space could be spent on carefully designed experiments showcasing the value of the flexibility provided with this framework.

The delivery of the whole paper is a bit too casual. And although this could certainly be a style and I'm not subtracting any points for it, the wording in many places could be much tighter. Phrases such as "measure theory side of things" (what things), "totally possible" and other long winded phrasings led me to this impression.

**Questions:**

The only experiment is a synthetic/toy image dataset (which is not a problem in itself) that is being reconstructed. And the reconstruction shows the homological hallmarks of a Klein bottle. But how is this surprising if the original image had the Klein bottle topology to begin with? If it can reconstruct pixelwise, then it also carries the topology.

Do the authors mean negative ELBO in Figure 5a? Since in the current way it is written in eqn (20), the higher the ELBO the better. Otherwise the Vanilla VAE's of 2-3-4 dimensions also perform better in the reconstruction loss.

What does being close in the bottleneck distance of persistence diagrams but not close in the reconstruction distance mean?


Below are not questions but some typos that I found.

Second line of equation (2) on line 100 is $(x -1 , 1-y)$. Otherwise $-y$ doesn't fall inside $[0,1]$ (this is how the authors realize the torus $T^2$). Or perhaps the authors want to use $-y$ and realizes the torus as $[-1,1]$.

In " where $t_\theta(x,y) = cos(\theta_1) x+sin(\theta_1) y$ is projection onto a line specified by $\theta_1$," the authors mean "length of projection" since the output is a number not a vector.

line 294 coveging --> covering

Line 720 $c_M$ --> $c(M)$ as used earlier. But also the notation $M, s_x, P, Q$ should all be defined in this section. I can see/guess that $M$ is a matching but $s_x$ is a mystery, possibly they are the $x$ and $y$ coordinates (birth/death) of a point on the persistence diagram and $|s_x - s_y|$ measures the lifetime of the unmatched pairs.  Anyways, I shouldn't have to guess.

Line 748 metricd --> metrics

Hard to read which line belongs to which legend in figure 5(b).

---

> ### Author Response · Authors · 2025-12-01
>
> Dear Reviewer,
> Thank you very much for your time and effort, and attention to our modest work!
> We've decided to withdraw it from ICLR, to further improve it.
> Again, thank you very much for your review, we highly appreciate all aspects of it!
> With all respect,
> Authors

---

### Official Review · Reviewer_jhRo · 2025-10-30

**Soundness:** 2
**Presentation:** 2
**Contribution:** 1
**Rating:** 2
**Confidence:** 4

**Summary:**

This paper generalizes the reparameterization trick for variational autoencoders (VAEs) to latent spaces with non-trivial topology by leveraging covering maps. The authors propose that when a covering map satisfies certain measure preservation properties, one can bound the KL-divergence between pushforward densities on the base latent manifold, making the KL term in the VAE's ELBO tractable despite topological complexity. As a proof of concept, the authors describe how one could construct a VAE with Klein bottle topology (KleinVAE). They suggest potential applications as weight priors for convolutional neural networks, citing connections to Gabor-Klein filters. Proof of concept experiments are given in the appendix.

**Strengths:**

- Reasonable central proposition: The idea of using covering maps to handle non-trivial topologies in VAE latent spaces is mathematically sound and represents a natural geometric approach to the problem.
- Change-of-volume-free formulation: The proposed method is interesting in that it does not require explicit computation of volume changes under the covering map, which could be advantageous in more complex settings.
- Novel topology: The Klein bottle example demonstrates that the framework can handle non-Lie group topologies, extending beyond the typical manifold structures considered in geometric deep learning.
- Clear mathematical framework: When viewed purely as a mathematical construction, the use of covering maps and the associated measure-theoretic properties is elegant.

**Weaknesses:**

- Insufficient experimentation: The toy experiments in the appendix are not on any kind of real data, which is insufficient for this venue.
- Limited application of theory: While the theory is more general, it is applied only to the flat geometry with $\mathbb{R}^n$ as base space and no volume change in the covering map, meaning the key advertised benefit (avoiding volume change computation) is never utilized.
- Unconvincing motivation: The relevance of Klein bottle topology to machine learning is poorly justified. The connection to Gabor-Klein filters appears contrived and its importance for practical applications remains unclear.
- Limited novelty: Covering maps have been used previously, such as in the case of the circle, and torus, as mentioned in the paper. Also other examples exist, such as Köhler et al. (2023) which uses covering spaces ($S^3$ covering $SO(3)$) for distribution modeling, although they do not use the proposed bound in this paper.
- Poor accessibility: The presentation assumes uncommon background in topology. Figure 2 and surrounding discussion are particularly unclear for a typical ML audience.
- Unaddressed limitations: The restriction to non-measure-increasing covering maps is not discussed, nor which topologies this excludes or whether it represents a fundamental limitation.

Ref: Köhler, J., Invernizzi, M., De Haan, P. & Noe, F.. (2023). Rigid Body Flows for Sampling Molecular Crystal Structures. ICML 2023

**Questions:**

- Tightness of bounds: Can the authors clarify under what conditions the bound from Proposition 1 becomes tight (i.e., achieves equality)? How loose is this bound in typical scenarios, and does this affect the quality of the variational approximation?

---

> ### Author Response · Authors · 2025-12-01
>
> Dear Reviewer,
> Thank you very much for your time and effort, and attention to our modest work!
> We've decided to withdraw it from ICLR, to further improve it.
> Again, thank you very much for your review, we highly appreciate all aspects of it!
> With all respect,
> Authors

---

### Official Review · Reviewer_wD2H · 2025-10-31

**Soundness:** 3
**Presentation:** 3
**Contribution:** 3
**Rating:** 4
**Confidence:** 2

**Summary:**

This paper extends the reparameterization trick in VAEs to latent spaces with non-trivial topology. Instead of assuming a Euclidean latent space, the authors allow latent variables to live on manifolds that are coverings of other manifolds, enabling reparameterization via measurable covering maps. Under certain measure-preservation conditions, they show that the KL divergence term in the ELBO remains tractable, even when the latent space has topological complexity.

The authors position their approach as related to—but distinct from—recent work on Lie-group-based reparameterization, arguing that Lie exponential maps can be seen as a special case of their more general framework. They introduce the concept of “reparameterization through a covering” and emphasize that covering maps need not be diffeomorphisms, thereby broadening the scope beyond smooth Lie-theoretic methods.

To demonstrate feasibility, the paper constructs a VAE with a Klein bottle latent space (“KleinVAE”) and trains it on a synthetic dataset. The results suggest that VAEs can successfully operate with such topology-structured latent spaces. The authors briefly discuss potential relevance for Bayesian learning and convolutional architectures, where non-Euclidean priors may be beneficial.

**Strengths:**

1. Novel theoretical contribution on VAE reparameterization. The paper proposes a new conceptual extension of the reparameterization trick to latent spaces with non-trivial topology, via covering maps. This is an under-explored direction in generative modeling and contributes to a deeper understanding of latent geometry in VAEs.

2. Broader generalization beyond Lie-group latent spaces. Prior work mostly focuses on Lie groups and smooth manifolds. This paper introduces a more general measurable-topology viewpoint, not requiring smoothness or Lie-group structure, broadening applicability.

3. Clear connection to manifold hypothesis. Links motivation to real-world data structure and the manifold hypothesis, situating the work in a meaningful theoretical context.

4. Strong mathematical grounding. Uses ideas from point-set topology, measure theory, and covering space theory, demonstrating conceptual maturity and rigor.

5. Constructive demonstration with Klein bottle latent space. The KleinVAE example provides a concrete proof-of-concept for learning with a non-trivial latent topology.

**Weaknesses:**

Limited empirical validation

1. The Klein bottle example is interesting but synthetic and small-scale. No experiments on real-world datasets or comparison to conventional topological latents (e.g., spherical VAEs, torus VAEs, Lie-group VAEs).

2. Lack of quantitative evaluation Claims about model benefits remain largely conceptual; empirical evidence appears primarily qualitative.

3. Assumptions may be restrictive or hard to verify. The measure-preservation condition for covering maps may not always hold or be trivial to check in practice.

4. Practical implications not fully explored. The paper discusses potential applications (e.g., Bayesian deep learning, convnets), but does not empirically demonstrate performance gains in these settings.

**Questions:**

Please see above for my questions.

---

> ### Author Response · Authors · 2025-12-01
>
> Dear Reviewer,
> Thank you very much for your time and effort, and attention to our modest work!
> We've decided to withdraw it from ICLR, to further improve it.
> Again, thank you very much for your review, we highly appreciate all aspects of it!
> With all respect,
> Authors

---

### Note · Authors · 2025-12-01

**Comment:**

We have decided to withdraw our work from ICLR-2026 to further improve it

**Withdrawal Confirmation:**

I have read and agree with the venue's withdrawal policy on behalf of myself and my co-authors.